# Measurements of Serum Mac-2-Binding Protein Glycosylation Isomer and Shear Wave Velocity in Health Checkups Are Useful in Screening for Non-Alcoholic Steatohepatitis

**DOI:** 10.3390/healthcare9050523

**Published:** 2021-04-29

**Authors:** Hideyuki Tamai, Jumpei Okamura, Takashi Ohoshi, Hisao Wakasaki

**Affiliations:** 1Department of Hepatology, Wakayama Rosai Hospital, 93-1 Kinomoto, Wakayama 640-8505, Japan; jup-okamura@wakayamah.johas.go.jp; 2Department of Clinical Examination, Wakayama Rosai Hospital, Wakayama 640-8505, Japan; tks-oohoshi@wakayamah.johas.go.jp; 3Department of Internal Medicine, Wakayama Rosai Hospital, Wakayama 640-8505, Japan; wakasaki@wakayama-med.ac.jp

**Keywords:** non-alcoholic steatohepatitis, Mac-2-binding protein glycosylation isomer, shear wave, screening

## Abstract

Liver-related mortality rates in patients with non-alcoholic steatohepatitis (NASH) increase with advancing liver fibrosis stage. The present study aimed to elucidate whether adding non-invasive liver fibrosis tests to a comprehensive health checkup system is useful for NASH screening. Both serum Mac-2-binding protein glycosylation isomer (M2BPGi) and point shear wave elastography (pSWE) using ultrasonography were performed for 483 health checkup subjects who consented to participate in this prospective study. Outcomes in positive subjects were surveyed 1 year later. Eighty-eight subjects (18%) showed positive results for at least one liver fibrosis test, with 63 subjects positive for pSWE, 33 subjects positive for M2BPGi, and 72 subjects showing no significant elevation of liver enzymes. The secondary consultation rate for positive subjects was 52% (46/88). However, as 15 of those 46 subjects visited a non-liver-specialist and could not undergo detailed examination, the secondary examination rate was only 35% (31/88). For the 31 subjects who received secondary examination, NASH was diagnosed in 14 subjects, other chronic liver disease (CLD) in 6 subjects, and no CLD in 11 subjects. Additional liver fibrosis tests using M2BPGi and pSWE appear useful in health checkups when screening for CLD, especially for NASH.

## 1. Introduction

Non-alcoholic fatty liver disease (NAFLD) is the most common cause of chronic liver disease around the world. Non-alcoholic steatohepatitis (NASH) represents 10–20% of NAFLD, and can progress to cirrhosis or hepatocellular carcinoma [1]. Most patients with NAFLD are obese, and many have type 2 diabetes, dyslipidemia and/or hypertension. As unhealthy lifestyles play key roles in the development and progression of NAFLD [2], this pathology is considered a component of metabolic syndrome [1]. As NASH patients remain asymptomatic until the liver deteriorates to a state of severe cirrhosis, early detection of NASH can be very difficult. Since serum transaminase levels are helpful in screening for NAFLD, NASH is often detected during routine health checkups or clinical visits for other diseases [1]. However, serum transaminase levels are not associated with NAFLD severity, and a substantial proportion of NAFLD patients show normal transaminase levels [3,4]. The stage of liver fibrosis represents the most useful predictor of mortality among NAFLD patients, and liver-related mortality rates increase exponentially as fibrosis stage advances, even at early stages [5]. Identification of NASH patients with liver fibrosis as early as possible in mass screenings such as health checkups is crucial, but must also lead to appropriate medical care with liver specialists.

Fortunately, a comprehensive health checkup system is available in Japan, as the so-called Ningen Dock. The Ningen Dock is widely performed all over Japan for early detection not only of cancers, but also lifestyle-related diseases. However, the basic tests in the complete medical checkup of the Ningen Dock do not currently include non-invasive tests to detect patients with liver fibrosis.

Many non-invasive methods for diagnosing liver fibrosis in NAFLD have already been proposed, such as serum fibrosis markers and scoring systems for predicting liver fibrosis, vibration-controlled transient elastography, ultrasound-based shear wave elastography, and magnetic resonance elastography [6]. NAFLD practice guidelines from the American Association for the Study of Liver Diseases recommends using scoring systems for predicting liver fibrosis such as fibrosis-4 (FIB-4) or NAFLD fibrosis score (NFS), transient elastography, and magnetic resonance elastography as first-line examinations to detect patients with advanced fibrosis [7]. However, tests fit for mass screening require not only accuracy, but also non-invasiveness, low cost, and wide availability. Transient elastography and magnetic resonance elastography are not widely available due to costs. Furthermore, FIB-4 and NFS have low specificities for advanced fibrosis in patients >65 years old, leading to an unacceptably high false-positive rate, because age is a confounding factor [8]. The basic tests of Ningen Dock already include blood tests and abdominal ultrasonography. Additional measurements of serum fibrosis markers and liver elastography using ultrasound would thus be easy to introduce with low cost and wide availability.

Mac-2-binding protein glycosylation isomer (M2BPGi) is a novel serum fibrosis marker identified in 2013. Mac-2-binding protein (M2BP) is a glycoprotein secreted from hepatic stellate cells, and specific glycan structures of M2BP change as liver fibrosis progresses [9]. The clinical use has rapidly increased in recent years mainly in Asia. M2BPGi values are unaffected by age, as opposed to scoring systems such as the FIB4 index or NFS [10]. This characteristic of M2BPGi is useful for mass screening.

The aim of the present study was to elucidate whether adding liver fibrosis tests of ultrasound-based shear wave elastography and M2BPGi to health checkup would prove useful for NASH screening.

## 2. Material and Methods

### 2.1. Subjects

This was a prospective cohort study about adding liver fibrosis tests to the Ningen Dock health checkup system. The population of this study was primary care. Between April 2018 and March 2019, among 583 health checkup subjects who underwent Ningen Dock in our hospital, 483 subjects consented to participate in the present study and subsequently underwent both M2BPGi and point shear wave elastography (pSWE) using ultrasound. Regardless of serum liver enzyme levels, subjects who showed positive results from at least one of the two liver fibrosis tests were recommended to undergo a detailed follow-up examination. The recommendation to receive detailed examinations at a liver-specializing institution on suspicion of advanced liver fibrosis or cirrhosis was mailed to positive subjects in addition to the report of Ningen Dock results. In addition, subjects showing significantly elevated levels of serum liver enzymes were also recommended to undergo detailed examination, as before. One year later, we surveyed outcomes in subjects for whom detailed examinations had been recommended. Final diagnosis was obtained from a consulted medical doctor.

### 2.2. Complete Medical Checkup Basic Tests

Basic test items of the one-day complete medical checkup for the 2018 version of the Japan Ningen Dock Society were performed as follows: (1) physical measurements (height, body weight, body mass index, and abdominal circumference); (2) physiological tests (blood pressure, electrocardiogram, heart rate, ocular fundus, intraocular pressure, visual acuity, audiometry, and spirometry); (3) chest X-ray lateral and anteroposterior views; (4) upper gastrointestinal series; (5) abdominal ultrasound; and (6) blood (total protein, albumin, creatinine, uric acid, high-density lipoprotein cholesterol, low-density lipoprotein cholesterol, triglycerides, aspartate aminotransferase (AST), alanine aminotransferase (ALT), γ-glutamyl transpeptidase (γGT), fasting blood glucose, hemoglobin A1c, and blood cell counts), urine, and stool tests. In accordance with criteria categories for the Japan Ningen Dock Society, subjects showing a significant elevation of any liver enzymes (defined as AST > 50 IU/L, ALT > 50 IU/L, or γGT > 100 IU/L) were recommended to receive liver-specialized medical care (detailed examination or treatment) as a category D.

### 2.3. Non-Invasive Liver Fibrosis Tests

Serum M2BPGi level was measured by chemiluminescence enzyme immunoassay (HISCL-5000; Sysmex, Kobe, Japan). A cut off index for M2BPGi ≥ 1.0 was defined as a positive result.

Shear wave velocity of the liver was also measured by pSWE using acoustic radiation force impulse on the same day as blood testing, using an ARIETTA850 ultrasound system (Hitachi, Tokyo, Japan). A median shear wave velocity from ≥10 measurements > 1.43 m/s was defined as positive [11].

### 2.4. Statistical Analysis

The secondary consultation rate was defined as the percentage of subjects who consulted a medical doctor among subjects referred for detailed examination. The secondary examination rate was defined as the percentage of subjects referred for detailed examinations who actually underwent detailed examinations. The NASH detection rate was defined as the percentage of subjects who were diagnosed as NASH among subjects referred for detailed examination. Secondary consultation rates and NASH detection rates between groups were compared using Fisher’s exact test or the χ^2^ test.

## 3. Results

### 3.1. Subject Characteristics

The 483 participants in this study comprised 243 males and 240 females, with a mean (±standard deviation) age of 55 ± 12 years. The relation of positive results among pSWM, M2BPGi and liver enzyme was shown in Figure 1. Of the 483 subjects enrolled in the present study, 88 subjects showed positive results for at least one of the two liver fibrosis tests, (positive rate, 18%). Of these 88 subjects with positive liver fibrosis tests, 16 subjects (18%) showed significantly elevated liver enzymes, and 72 subjects (82%) did not. The 88 subjects with positive liver fibrosis tests comprised 63 subjects with positive pSWM, 33 subjects with positive M2BPGi. Positive rates for pSWM and M2BPGi were 13% (63/483) and 7% (33/483), respectively. Eight subjects (9%) showed positive results for both pSWM and M2BPGi. Among the total 483 subjects, 64 subjects (13%) displayed significantly elevated liver enzymes. Of the 395 subjects with negative results for both liver fibrosis tests, 48 subjects showed significant elevation of liver enzymes, and were recommended to undergo detailed examination.

### 3.2. Outcomes after Follow-Up Survey

The flow of subjects through this study is shown in Figure 2. The secondary consultation rate for subjects with a positive liver fibrosis tests was 52% (46/88). However, 15 (33%) of the 46 subjects who had a secondary consultation visited a non-liver specialist. and thus could not undergo detailed examination. These patients were referred by the doctor for follow-up. The secondary examination rate was 35% (31/88). The 31 subjects who received a detailed examination were diagnosed with NASH in 14 cases (45%), alcoholic steatohepatitis (ASH) in 3 (10%), primary biliary cholangitis (PBC) in 2 (6%), and autoimmune hepatitis (AIH) in 1 (3%). The remaining 11 subjects (35%) were not diagnosed with any chronic liver disease (CLD). The 31 subjects who received a detailed examination comprised 16 subjects with positive pSWM and 15 subjects with positive M2BPGi. Of the 16 subjects with positive pSWM, the diagnosis was NASH in 7 subjects (44%), ASH in 2 subjects (13%), PBC in 1 subject (6%), AIH in 1 subject (6%), and no CLD in 5 subjects (31%). Of the 15 subjects with positive M2BPGi, the diagnosis was NASH in 7 subjects (47%), ASH in 2 subjects (13%), PBC in 1 subject (7%), and no CLD in 5 subjects (33%).

Among the 48 subjects recommended for a detailed examination based on liver enzyme levels, 16 subjects visited a medical institution to receive a detailed examination. The secondary consultation rate of subjects with significantly elevated liver enzyme levels was 33% (16/48). Of the 16 subjects, 6 subjects (37%) consulted a liver specialist, and 10 subjects (63%) consulted a non-specialist. Those 10 subjects could not receive detailed examination, and were referred for follow-up. The secondary examination rate was thus 13% (6/48). For these 6 subjects who consulted a liver-specialist, the diagnosis was NASH in 4 subjects, and ASH in 2 subjects.

The secondary consultation rate was significantly higher for the liver fibrosis group (52%) than for the elevated liver enzyme group (33%, *p* = 0.034). The secondary examination rate was also significantly higher for the liver fibrosis group (35%) than for the elevated liver enzyme group (13%, *p* = 0.005). Finally, NASH detection rates were 16% (14/88) in the liver fibrosis group and 8% (4/48) in the elevated liver enzyme group (*p* = 0.292).

## 4. Discussion

This prospective cohort study evaluated the utility of adding a non-invasive liver fibrosis tests to the comprehensive Ningen Dock health checkup system. Additional non-invasive liver fibrosis tests were found to identify a greater number of CLD patients, especially for NASH, among health checkup subjects than liver function tests alone. However, the present study also revealed some problems that needed to be addressed on the path to appropriate medical care.

Currently available non-invasive liver fibrosis tests can be classified into two types: biological methods based on the quantification of biomarkers in serum samples, or physical methods based on measurement of liver stiffness corresponding to intrinsic physical properties of liver parenchyma. These methods are based on different rationales and are complementary [12]. Loomba et al. indicated in a review article that combining serum-based tests with elastography techniques measuring liver stiffness increases diagnostic accuracy and can be used as a screening test [13]. In the present study, the rate of patients showing double-positive results for pSWM and M2BPGi was only 9%, and most positive subjects were positive for only a single test. Addition of these two different non-invasive liver fibrosis tests to health checkups thus appears reasonable and more useful in screening for subjects with liver fibrosis than using any one of the two tests. Furthermore, as 82% of patients with positive results from liver fibrosis tests showed no significant elevation of liver enzymes in the present study, these subjects would not have been identified using only the basic tests of the Ningen Dock. From this perspective, additional liver fibrosis tests would play an important role in screening for CLD.

As follow-up surveys in the present study, both secondary consultation rate and secondary examination rate were significantly higher for subjects with positive results from liver fibrosis tests than for subjects with significantly elevated liver enzymes. This might be attributable to the greater weight given to suspicion of advanced liver fibrosis or cirrhosis than to elevated liver enzymes. Additional liver fibrosis tests at the health checkup were thus more useful for leading CLD patients to appropriate medical care than liver enzyme tests alone. However, regardless of recommendations to consult a liver specialist, not all positive subjects consulted a suitable specialist. No subjects who consulted a non-specialist were able to receive detailed examinations, and all were instead recommended for follow-up. This represents a serious problem in achieving appropriate medical care. To address this issue, education about CLD and regional cooperation between liver specialists and non-specialists are needed.

In the present study, all subjects who consulted liver specialists were able to receive appropriate specialized medical care. Detailed examinations suggested that most CLD was caused by NASH or ASH associated with lifestyle-related disease. Additional liver fibrosis tests were thus useful in screening for lifestyle-related chronic liver disease among health checkup subjects. In addition, some patients with PBC or AIH could also be identified from health checkups. As liver fibrosis tests are not etiologically specific, adding further liver fibrosis tests to health checkups may be also useful in mass screening for CLD. However, approximately one-third of positive subjects who underwent further detailed examinations were diagnosed as not showing CLD in the present study. This relatively high false-positive rate might depend on the accuracy, cutoff level, and/or confounding factors for liver fibrosis tests. Cassinotto et al. demonstrated that the area under the receiver operating curve (AUC) of pSWE was 0.77 for diagnosing fibrosis stage ≥F2, 0.84 for ≥F3, and 0.84 for F4, respectively, and the predictability of ≥F2 was lower than that of ≥F3 [14]. In addition, they also indicated that unreliable results of pSWE were shown in 18.2% of patients. Meta-analyses of pSWE in patients with CLD have reported diagnostic accuracies of 89–91% for advanced fibrosis (≥F3) and 92–93% for cirrhosis, with cutoffs ranging from 1.55–1.61 m/s for advanced fibrosis (≥F3) and 1.80–1.87 m/s for cirrhosis [15,16]. In the present study, as the cutoff level was set at 1.43 m/s, the accuracy might have been relatively low. On the other hand, Tamaki et al. indicated that positive predictive values for significant fibrosis (≥F2) with cutoff M2BPGi levels of ≥1.0, ≥1.1, ≥1.2, ≥1.3, ≥1.4, and ≥1.5 were 29.2%, 36.4%, 43.5%, 42.9%, 62.5%, and 71.4%, respectively [17]. Furthermore, as the cutoff M2BPGi level for predicting advanced fibrosis (≥F3) in NAFLD was lower than that for predicting advanced fibrosis in chronic hepatitis C, the underlying liver disease must be considered when interpreting M2BPGi results [9]. As most data for M2BPGi cutoffs are based on patients attending liver clinics, little data has been obtained from health checkup subjects. Nah et al. indicated in a study of selected health checkup subjects that the AUCs of cutoff M2BPGi levels in screening for fibrosis stages ≥F1, ≥F2, and ≥F3 were 0.591, 0.698, and 0.853, respectively, with a threshold of 0.75 considered optimal to distinguish advanced fibrosis (≥F3) with 80.0% sensitivity, 77.9% specificity, and 98.9% negative predictive value. They also demonstrated in same article that the AUC for distinguishing advanced fibrosis (≥F3) was better for M2BPGi than for aspartate aminotransferase-to-platelet ratio or FIB-4 [18]. Future studies should clarify appropriate cutoffs for mass screening in health checkup subjects.

Some limitations to the present study need to be considered. First, this prospective study only involved a single center and thus may have included various selection biases, and the number of subjects was small for mass screening. A larger-scale multicenter study is needed to validate our results. Second, although FIB-4 and NFS are easily available from standard laboratory data, whether the combination of M2BPGi and pSWE is superior to these liver fibrosis scoring systems in mass screening could not be evaluated in the present study. Of course, FIB-4 and NFS could be calculated using the data of our health check-up system for this cohort. However, FIB-4 and NFS are not still used in the “Ningen Dock” Japanese health check-up system, and these markers do not have enough evidence in mass screening. Comparative studies with FIB-4 or NFS and single serum fibrosis markers such as M2BPGi should be performed in the future. Third, the present study cannot provide the predictability, such as receiver operating characteristic curves, sensitivity, specificity, positive predictive value, negative predictive value, and accuracy, to validate the applicability of M2BPGi and pSWE, because subjects with negative results of these tests were not surveyed. Fourth, the cost effectiveness of additional combination tests of M2BPGi and pSWE was not evaluated in the present study. It is necessary to evaluate the cost effectiveness of these tests in the future. 

## 5. Conclusions

Adding M2BPGi and pSWE to the Ningen Dock comprehensive health checkup system as liver fibrosis tests appears useful in screening for chronic liver disease, especially NASH. Asymptomatic CLD subjects identified in screening examinations can be directed to appropriate medical care, taking efforts to optimize the secondary examination rate.

## Figures and Tables

**Figure 1 healthcare-09-00523-f001:**
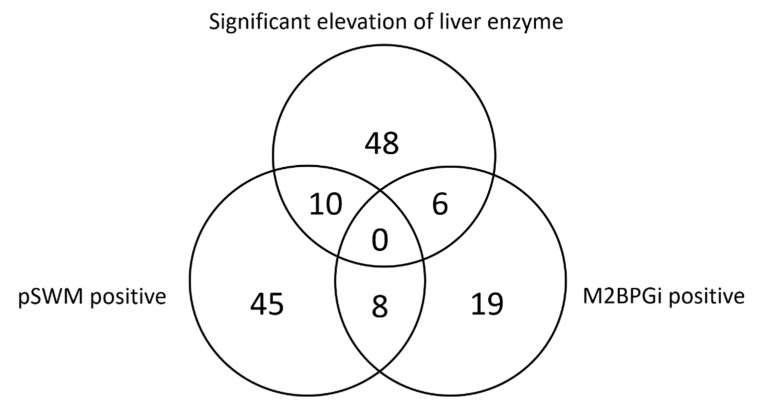
Relation of positive results among additional liver fibrosis tests and liver enzyme. M2BPGi, Mac-2-binding protein glycosylation isomer; pSWE, point shear wave elastography.

**Figure 2 healthcare-09-00523-f002:**
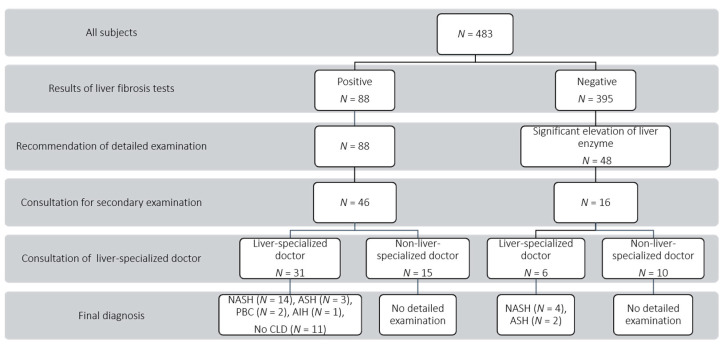
Flow chart for follow-up survey. NASH, non-alcoholic steatohepatitis; ASH, alcoholic steatohepatitis; PBC, primary biliary cholangitis; AIH, autoimmune hepatitis; CLD, chronic liver disease.

## Data Availability

The database for the current study is available from the corresponding author on reasonable request.

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
