# Peer review of "Measurements of Serum Mac-2-Binding Protein Glycosylation Isomer and Shear Wave Velocity in Health Checkups Are Useful in Screening for Non-Alcoholic Steatohepatitis"

_healthcare, 2021, doi:10.3390/healthcare9050523_

Round 1

Reviewer 1 Report

Tamai et al describe in this manuscript a combined method comprising of serum Mac-2-binding protein glycosylation 2 isomer and shear wave velocity to screen for nonalcoholic steatohepatitis during health checkups.

As diagnosis of this condition is extremely important, this study is novel and of relevance.

I have only minor comments:

A bit more description of the MAC2 proteins is needed for the reader to really capture the essence of the study.

In the results section, a table containing the different percentages is needed to help the reader through this section.

line 184-185: sentence is not very clear; please rephrase it.

Author Response

Reviewer 1

Comments and Suggestions for Authors

Tamai et al describe in this manuscript a combined method comprising of serum Mac-2-binding protein glycosylation 2 isomer and shear wave velocity to screen for nonalcoholic steatohepatitis during health checkups.

As diagnosis of this condition is extremely important, this study is novel and of relevance.

I have only minor comments:

A bit more description of the MAC2 proteins is needed for the reader to really capture the essence of the study.

→Given the reviewer’s comment, the description of M2BPGi was added in line 66-69.

In the results section, a table containing the different percentages is needed to help the reader through this section.

→Given the reviewer’s comment, The relation of positive results among pSWM, M2BPGi and liver enzyme was shown in Figure 1 to make the results easy to understand.

line 184-185: sentence is not very clear; please rephrase it.

→Given the reviewer’s comment, the sentence of line 184-185 was rephrased (line 190-192).

Reviewer 2 Report

Thank you for submitting this paper.

Regarding the design of the study: it would be very useful to know whether this population was primary or secondary care - hence the validation and applicability of the proposed test. 

Also, please clarify as NASH detection rates were defined, as there is no mentioning of liver biopsies, which is the only mean to diagnose NASH.

In terms of methods - the results should be implemented with ROC curves, NPV and PPV, accuracy as these parameters provide an idea of the applicability of the test. Also, given the size of the population, it would be good to split it into a derivation and validation cohort. Given the lack of the above-mentioned results, the interpratation of the tests is limited.

Author Response

Reviewer 2

Thank you for submitting this paper.

Regarding the design of the study: it would be very useful to know whether this population was primary or secondary care - hence the validation and applicability of the proposed test. 

→Given the reviewer’s comment, the sentence “ The population of this study was primary care” was inserted in line 78.

Also, please clarify as NASH detection rates were defined, as there is no mentioning of liver biopsies, which is the only mean to diagnose NASH.

→Given the reviewer’s comment, NASH detection rate was defined in line 117-119. The methods of NASH diagnosis were unknown because of hearing investigation. So, the sentence “Final diagnosis was obtained with hearing investigation from consulted medical doctor.” was added in line 89-90.

In terms of methods - the results should be implemented with ROC curves, NPV and PPV, accuracy as these parameters provide an idea of the applicability of the test. Also, given the size of the population, it would be good to split it into a derivation and validation cohort. Given the lack of the above-mentioned results, the interpratation of the tests is limited.

→We agree your comments. However, the predictability of M2BPGi and pSWM cannot be evaluated, because we did not survey subjects with negative results. So, this point is a limitation of this study. We added this limitation in line 241-245.

Reviewer 3 Report

Dear Authors

Congratulations for this interesting study. The incorporation of this novel marker ,M2BPGi in conjunction with pSWE  as mass health screening test in your country health check up system, is a smart idea. Of course ,as you have pointed out, the number of enrolled subjects for mass screening health test is small. It is impressive from your results that only 9% of 88 (out of 483 subjects) were positive for both liver fibrosis test, M2BPGi and pSWE . This finding creates a question. Is it cost effective and useful to use as screening test both of these two markers? What could be the accuracy if you have used the one of these two markers? Have you analyzed separately for M2BPGi and pSWE the predictability or the accuracy for diagnosis  Chronic Liver Disease(CLD)? Moreover,because you have  the data of your health check up system for this cohort, you could see the predictability  of M2BPGi in comparison with FIB-4 or/and NFS of CLD or the accuracy in diagnosis liver fibrosis?

Author Response

Reviewer 3

Congratulations for this interesting study. The incorporation of this novel marker ,M2BPGi in conjunction with pSWE  as mass health screening test in your country health check up system, is a smart idea. Of course ,as you have pointed out, the number of enrolled subjects for mass screening health test is small. It is impressive from your results that only 9% of 88 (out of 483 subjects) were positive for both liver fibrosis test, M2BPGi and pSWE . This finding creates a question.

Is it cost effective and useful to use as screening test both of these two markers?

→Cost effectiveness cannot be evaluated in the present study. This point is a limitation of this study. We added this limitation in line 245-247.

What could be the accuracy if you have used the one of these two markers?

Have you analyzed separately for M2BPGi and pSWE the predictability or the accuracy for diagnosis  Chronic Liver Disease(CLD)?

Moreover, because you have the data of your health check up system for this cohort, you could see the predictability of M2BPGi in comparison with FIB-4 or/and NFS of CLD or the accuracy in diagnosis liver fibrosis?

→We did not survey subjects with negative results of M2BPGi and pSWM. So, predictability of these fibrosis tests cannot be evaluated. This point is also a limitation of this study. We added this limitation in line 241-245.

Round 2

Reviewer 2 Report

all my comments have been addressed